Cytokine response during non-cerebral and cerebral malaria: evidence of a failure to control inflammation as a cause of death in African adults

Dieye Yakhya 1 y.dieye@laposte.net
Mbengue Babacar 2 3
Dagamajalu Shobha 4
Fall Mouhamadou Mansour 5
Loke Mun Fai 4
Nguer Cheikh Momar 6
Thiam Alassane 3
Vadivelu Jamuna 4
Dieye Alioune 2 3
1 Vice-Chancellor’s Office, University of Malaya , Kuala Lumpur , Malaysia
2 Département d’Immunologie, Faculté de Médicine, de Pharmacie et d’Odontostomatologie, Université Cheikh Anta Diop de Dakar , Dakar , Sénégal
3 Unité d’Immunogénétique, Institut Pasteur de Dakar , Dakar , Sénégal
4 Department of Medical Microbiology, Faculty of Medicine, University of Malaya , Kuala Lumpur , Malaysia
5 Service de Réanimation, Hôpital Principal de Dakar , Dakar , Sénégal
6 Département Génie Chimique et Biologie Appliquée, École Supérieure Polytechnique, Université Cheikh Anta Diop de Dakar , Dakar , Sénégal
Braga Erika
Electronic publication date: 2016 May 2
Publication date: 2016
Volume: 4
Electronic Location ID: e1965
Received 2015 Dec 7; Accepted 2016 Apr 2
Copyright: ©2016 Dieye et al.
Copyright year: 2016
Copyright holder: Dieye et al.
License: This is an open access article distributed under the terms of the Creative Commons Attribution License, which permits unrestricted use, distribution, reproduction and adaptation in any medium and for any purpose provided that it is properly attributed. For attribution, the original author(s), title, publication source (PeerJ) and either DOI or URL of the article must be cited.
License URL: https://creativecommons.org/licenses/by/4.0/

Keywords: Malaria, Cerebral, Plasmodium falciparum, Cytokine, Inflammation

Funding: University of Malaya-Ministry of Education (UM-MoE) High Impact Research (HIR) Grant UM.C/625/1/HIR/MoE/CHAN/13/6 This study was supported by University of Malaya-Ministry of Education (UM-MoE) High Impact Research (HIR) Grant UM.C/625/1/HIR/MoE/CHAN/13/6 (account no. H-50001-A000034). The funders had no role in study design, data collection and analysis, decision to publish, or preparation of the manuscript.

==============================
Background. With 214 million cases and 438,000 deaths in 2015, malaria remains one of the deadliest infectious diseases in tropical countries. Several species of the protozoan Plasmodium cause malaria. However, almost all the fatalities are due to Plasmodium falciparum, a species responsible for the severest cases including cerebral malaria. Immune response to Plasmodium falciparum infection is mediated by the production of pro-inflammatory cytokines, chemokines and growth factors whose actions are crucial for the control of the parasites. Following this response, the induction of anti-inflammatory immune mediators downregulates the inflammation thus preventing its adverse effects such as damages to various organs and death.

Methods. We performed a retrospective, nonprobability sampling study using clinical data and sera samples from patients, mainly adults, suffering of non-cerebral or cerebral malaria in Dakar, Sénégal. Healthy individuals residing in the same area were included as controls. We measured the serum levels of 29 biomarkers including growth factors, chemokines, inflammatory and anti-inflammatory cytokines.

Results. We found an induction of both pro- and anti-inflammatory immune mediators during malaria. The levels of pro-inflammatory biomarkers were higher in the cerebral malaria than in the non-cerebral malaria patients. In contrast, the concentrations of anti-inflammatory cytokines were comparable in these two groups or lower in CM patients. Additionally, four pro-inflammatory biomarkers were significantly increased in the deceased of cerebral malaria compared to the survivors. Regarding organ damage, kidney failure was significantly associated with death in adults suffering of cerebral malaria.

Conclusions. Our results suggest that a poorly controlled inflammatory response determines a bad outcome in African adults suffering of cerebral malaria.

Introduction

Despite a decade of sustained efforts that have substantially reduced mortality and morbidity due to malaria, this disease continues to represent an important health concern in tropical countries (White et al., 2014). According to the World Health Organization (WHO), there were 214 million cases of malaria worldwide in 2015, which resulted in 438,000 deaths (WHO, 2015). Ninety percent of the victims were from Africa, 74% being children under five years of age. Malaria is endemic in many sub-Saharan African countries. However, there are disparities between (WHO, 2015) and even within countries (Espie et al., 2015) regarding the transmission of the disease. In many rural areas where the local environment favors the development of the mosquito vector and its interactions with humans, transmission of malaria is high and perennial (Trape et al., 2014). In contrast, in other areas including urban zones, the transmission of malaria is low to moderate and seasonal (White et al., 2014). Individuals living in regions of high and stable transmission progressively acquire immunity after experiencing and surviving to several infections (Olliaro, 2008). This immunity protects against severe, life-threatening cases of malaria but does not confer a sterile protection (Doolan, Dobano & Baird, 2009). In these areas, clinical malaria occurs in young children while healthy carriage of the parasite is common in adults. Adults who die of malaria typically are pregnant women or non-immune individuals from low transmission zones.

There was a decline of 18% and 48% of global malaria cases and deaths respectively between 2000 and 2015. This success was primarily due to a drastic reduction of malaria transmission by widespread use of insecticide-treated bednets and the availability of artemisinin-based treatments (Bhatt et al., 2015; White et al., 2014). The decline of malaria is expected to continue with the support of the WHO Global Technical Strategy for malaria 2016–2030 that aims to reduce its global incidence and mortality by at least 90% by 2030 (WHO, 2015). However, a continuous decrease of malaria prevalence and especially a sustained reduction of transmission in currently holoendemic zones may result, in the future, in an increase of the number of adults susceptible to severe cases. Such an unwanted but possible scenario potentially represents a future public health concern in sub-Saharan African countries. Several species of the protozoan Plasmodium cause malaria. However, almost all the deaths are due to P. falciparum, a species that causes the severest cases including cerebral malaria (Storm & Craig, 2014). In response to P. falciparum infection, a robust immune inflammatory response takes place. An important component of this response is the production of inflammatory immune mediators whose actions are crucial for the control of the parasites (Deloron et al., 1994; Lyke et al., 2004; Sarthou et al., 1997). This inflammatory response is rapidly followed by the production of anti-inflammatory cytokines that downregulate the inflammation preventing detrimental immune reactions (Kurtzhals et al., 1998; Peyron et al., 1994; Walther et al., 2005). Therefore, the immune response to P. falciparum infection includes a subtle balance of pro- and anti-inflammatory immune mediators (Crompton et al., 2014; Frosch & John, 2012). A rupture of this balance is at the basis of the events that lead to organ damage and death (Crompton et al., 2014). The pathogenesis of severe malaria and its associated mortality have been widely studied in children. In contrast, there are less investigations that addressed these aspects of the disease in adults, in particular from Africa (Olliaro, 2008). In this study, we performed a retrospective analysis of the available clinical data and of the immune response of malaria patients, mainly adults, admitted at the Hôpital Principal de Dakar, Sénégal. Malaria is endemic in several areas in Sénégal. However, the capital city Dakar and its surroundings constitute a zone of low prevalence of malaria with a seasonal transmission. We report the analysis of the serum levels of cytokines, chemokines and growth factors in control individuals and in patients suffering of non-cerebral (NCM) or cerebral malaria (CM). All the CM patients were adults and included deceased and survivors enabling to gain insights into the effect of the analysed biomarkers in the outcome of the disease.

Materials and Methods

Study population, ethics, consent and permissions

This study was performed on serum samples from patients diagnosed with malaria at the Hôpital Principal de Dakar, Sénégal between October 2012 and December 2014 (Torrentino-Madamet et al., 2014). The samples were taken after written consents from the patients or their accompanying family members. The controls corresponded to samples obtained from healthy volunteers residing in Dakar. This study was approved by the Université Cheikh Anta Diop de Dakar’s institutional research ethics committee (Protocol No 001/2015/CER/UCAD). Venous blood samples were collected in Vacutainer® ACD tubes (Becton Dickinson, Rutherford, NJ, USA) prior to patient treatment. Plasmodium presence and density in blood samples were determined by microscopic examination of thin blood smears stained with a 10% May-Grünwald Giemsa solution (SigmaR, St-Louis, MO, USA). P. falciparum was the only species found. Blood parameters were determined at the hospital’s clinical laboratory. The following criteria were used for enrollment into the two groups of malaria patients. Life-threatening CM was defined following the WHO criteria as the presence of P. falciparum in blood smears accompanied by a coma with no other cause of cerebral symptoms. NCM cases were defined by fever and presence of P. falciparum in blood smear, without other infections or symptoms of severe malaria as defined by the WHO (WHO, 2000). CM patients were treated according to a protocol based on the Senegalese national recommendations that consisted of intramuscular administrations of 20 mg/kg quinine every eight hours. NCM patients were treated with oral administration of 20 mg/kg of artesunate derivates or quinine. Secondary samples analyzed in this study corresponded to blood taken from survivors of CM (14 individuals) before patient release from the hospital (1–15 days after admission).

Biomarker measurement

Serum biomarkers were measured using a Milliplex MAP kit for human cytokine/ chemokine magnetic bead panel (catalogue # HCYTMAG-60K-PX29; EMD Millipore Corporation, Billerica, MA, USA) according to the recommendations of the manufacturer. The levels of 29 biomarkers were measured in each sample including interleukin (IL)-1α, IL-1β, IL-1RA, IL-2, IL-3, IL-4, IL-5, IL-6, IL-7, IL-8, IL-10, IL-12 (p40), IL-12 (p70), IL-13, IL-15, IL-17, interferon (IFN)α2, IFNγ, IFN-inducible protein 10 (IP-10, CXCL-10), epidermal growth factor (EGF), eotaxin, granulocyte colony-stimulating factor (G-CSF), granulocyte-macrophage colony-stimulating factor (GM-CSF), tumor necrosis factor (TNF)α, TNFβ, monocyte chemotactic protein (MCP)-1, macrophage inflammatory protein (MIP)-1α, MIP-1β, and vascular endothelial growth factor (VEGF). The measurements were performed in 25 ul of undiluted serum samples on one 96 well plate. Each well contained fluorescent-coded magnetic microbeads coated with analyte-specific capture antibodies to simultaneously measure the biomarkers in a specimen. Seven standards and two quality-control (QC) samples were included and measured in duplicate. The QC samples corresponded to mixtures with two values (high and low) for each biomarker. After the capture of the biomarkers, the beads were washed, incubated with biotinylated antibodies and then with streptavidin-PE. Excitation and fluorescence acquisition from the beads were performed using a Luminex 200™ equipped with an xPONENT™ software version 3.1 (Luminex, Austin, TX, USA) that calculated the concentrations of the biomarkers by extrapolating the mean fluorescence intensity (MFI) to a 5-parameter weighted logistic regression curve from the standards. Any measurement below the detection limit was given a value of 0 for the corresponding analyte. For most of the biomarkers, the majority of the samples had detectable values. For IL-2, IL-3, IL-4, IL12p40, IL-13 and TNFβ, the small number of samples with detectable MFI did not permit meaningful statistical analyses. These biomarkers were excluded from the statistical analyses. The MFI for one G-CSF and two IL-1RA samples were above the value of the highest standard whose concentration was 10,000 pg/ml. These samples were arbitrarily assigned concentrations above the highest standard. The G-CSF sample was assigned a concentration of 11,000 pg/ml. The two IL-1RA samples were attributed concentrations of 11,000 pg/ml and 12,000 pg/ml respectively according to the values of their MFI. Treating the samples above the highest standard in this way, rather than excluding them, allowed to take them into account when determining the median values for G-CSF and IL-1RA.

Statistical analyses

The statistical analyses were carried out using the IBM SPSS 22.0 software. Non-parametric tests were used to compare the levels of biomarker and their correlation with other variables across different groups. Mann–Whitney U test and Kruskal–Wallis one-way ANOVA were used to compare data across two and three groups respectively. Wilcoxon matched pairs signed rank test was used to compare biomarker levels in sera from CM patients at admission and at their release from the hospital. Correlation tests were performed using Spearman’s Rho rank test. Pearson Chi-Square was used to test association of organ failure with outcome in CM patients. Benjamini–Hochberg correction was used for multiple testing adjustment. For all the statistical analyses, a p value < 0.05 was considered as significant except when multiple testing adjustment was used, in which cases significant p values depended on the critical values from the Benjamini–Hochberg correction. The biomarker profiles were determined as previously described (Da Costa et al., 2014). In brief, the median value in the global population (CT + NCM + CM) was calculated for each biomarker and used as a cut-off to determine the percentage of individuals that had “high” (above the median value) and “low” (below the median value) levels of biomarker in the CT, NCM and CM groups. An ascendant biomarker profile was then constructed in the CT group by assembling the biomarkers from the one having the smallest percentage of high producers to the one having the largest. The resulting ascendant curve was used as a reference to visualize the variation of the percentage of high producers of biomarkers in the other groups (Fig. 1). In addition to showing the differences in the percentage of high producers (descriptive statistics), the biomarker profiles also indicate the analytes (hatched bars) for which there was a significant difference in adjusted Mann–Whitney U pairwise comparison (inferential statistics).

Figure 1 Serum levels of immune mediators during malaria.

The levels of 29 biomarkers were measured in control subjects (CT) and in non-cerebral (NCM) and cerebral (CM) malaria patients. The median value of each cytokine in the global population (CT + NCM + CM) was used as a cut-off value to determine the percentage of “high” (above median) biomarker producer individuals in each group. The ascendant biomarker profile of the CT (A) was determined and the resulting curve used as a reference to visualize the difference in the proportion of high biomarker producers with the NCM (B) and CM (C) groups. Hatched bars represent biomarkers for which there is a significant difference in Mann–Whitney U pairwise comparison with the CT reference group after Benjamini–Hochberg multiple test adjustment.

Results

Study population and clinical data

We performed a retrospective, nonprobability sampling study using sera samples from healthy individuals and from patients admitted at the Hôpital Principal de Dakar, Sénégal. The cohort included 17 and 27 subjects diagnosed with NCM and CM respectively, and 18 healthy controls (CT) (Table 1). The three groups of individuals were comparable in age and gender, and were mainly composed of adults (Table 1). Subjects below 15 years of age included six children aged 5–13 diagnosed with NCM. Several clinical and blood parameters existed but were not recorded for all the individuals preventing reliable statistical analyses. Available data showed, as expected, hemoglobin levels comparable in the CT and NCM individuals while significantly lower in the CM patients (Table 1). Additionally, parasitemia was comparable between the NCM and CM groups (Table 1). Regarding organ defect in the CM group, kidney failure was the most frequent (13/27) followed by liver, hematologic and respiratory malfunction, while hemodynamic failure was rare (Table 1). All the NCM patients were successfully treated, while 9/27 CM subjects died.

Table 1 Demographic, clinical, and disease outcome data.

		CT	NCM	CM	Total	
Gender	Male	11	11	22	44	
Female	7	6	5	17	
Age	Range	23–57	5–74	15–80	5–80	
Median	28.5	18	26	26	
HB	Normal	18	17	11	46	
Low	0	0	16	16	
Parasitemia	Median	0	520	1,452	1,022a	
IQR	0	2,552	9,981	7,654a	
Outcome	Survived	18	17	18	53	
Deceased	0	0	9	9	
Organ failure	Neurological	0	0	27	27	
Respiratory	0	0	6	6	
Kidney	0	0	13	13	
Liver	0	0	8	8	
Hematologic	0	0	7	7	
Hemodynamic	0	0	2	2	
Notes.

CT Control Individuals

NCM Non-cerebral Malaria Patients

CM Cerebral Malaria Patients

HB Hemoglobin (Low, <100 g/L; Normal, >100 g/L)

IQR Inter Quartile Range

a Value for NCM + CM.

Figure 2 Serum biomarker levels in control individuals and in non-cerebral and cerebral malaria patients.

Biomarkers that significantly differed across the three groups in Kruskal–Wallis test after Benjamini–Hochberg adjustment are shown. Box plots represent medians with 25th and 75th percentiles, bars 10th and 90th percentiles, and dots outliers for biomarker concentrations. P, p[i] values in Kruskal–Wallis tests. C, critical values in Benjamini–Hochberg correction.

Levels of inflammatory but not of anti-inflammatory biomarkers were higher in CM than in NCM patients

To analyze the production of immune mediators during malaria, we measured the serum levels of 29 biomarkers including growth factors, chemokines, inflammatory and anti-inflammatory cytokines. We determined the ascendant biomarker profile of the CT group as previously described (Da Costa et al., 2014) and plotted the resulting curve on the profiles of the NCM and CM patients (Fig. 1). The biomarker profiles display comparison of the proportion of individuals with levels of analytes above the global median between the controls and the malaria patients (Figs. 1B and 1C). Additionally, analytes that significantly differ in Mann–Whitney U pairwise comparison after Benjamini–Hochberg multiple testing adjustment are shown in the biomarker profiles (Figs. 1B and 1C, hatched bars). Several analytes were significantly higher in malaria patients (NCM and/or CM) than in CT individuals (Figs. 1 and 2). These biomarkers included most of the pro-inflammatory cytokines and chemokines tested (IL-1α, IL-6, IL-8, IL-12p70, IL-15, IL-17A, IP-10, TNFα, IFNα2, IFNγ, MIP-1α, MIP-1β and MCP-1) and the anti-inflammatory cytokines IL-10 and IL-1RA. The induction of both inflammatory and anti-inflammatory immune mediators that aims to respond to the infection while controlling the level of inflammation in order to prevent damages to host organs has been well documented in malaria patients (Frosch & John, 2012). Also, we observed an induction of Th1 (IL-12, IFNγ and TNFα) and Th2 (IL-10) biomarkers in both NCM and CM patients. Since only the control group included children, we performed pairwise comparisons after removing subjects below 15 years of age. We did not find differences in the analytes that significantly differed whether children were included or not except for IFNα2 that lost significance in the CT/NCM comparison (not shown). Besides individual biomarkers, several analytes were significantly positively correlated consistent with the immune response during malaria that mobilizes several cytokines, chemokines and growth factors (File S1). Next, we analyzed the difference of cytokine levels between the NCM and CM patients. Most of the biomarkers induced during malaria were higher in CM than in the NCM individuals (Fig. 3). However, after Benjamini–Hochberg adjustment for multiple tests (Table 2), only the levels of the pro-inflammatory IL-6 and IL-8, and of IL-1RA, an antagonist of IL-1 reached statistical significance (Fig. 3 and Table 2). These results indicate an inflammatory response of higher magnitude in CM compared to NCM patients as previously mentioned in several reports (Crompton et al., 2014). In contrast, the level of IL-10 was not significantly different between NCM and CM patients (Fig. 3). Additionally, the level of IL-5, a Th2 anti-inflammatory cytokine, was significantly lower in CM than in NCM patients (Fig. 3 and Table 2) as was the ratios of IL-5 to the pro-inflammatory biomarkers TNFα, IP-10 and IL-8 (not shown). These results suggest that the level of anti-inflammatory response did not match the strength of the inflammatory cytokine response in the CM patients.

Figure 3 Levels of inflammatory immune mediators are higher in cerebral than in non-cerebral malaria patients.

The ascendant biomarker profile curve of the NCM (line) was plotted on the CM graph (bars) to visualize the difference in the proportion of high biomarker producers. Hatched bars represent biomarkers for which there is a significant difference in Mann–Whitney U pairwise comparison between the two groups after Benjamini–Hochberg multiple test adjustment.

Table 2 Biomarkers significantly differing between non-cerebral and cerebral malaria patients after multiple testing adjustment.

Biomarkers	MW P values	BH critical values	Significance	
TNFαa	0.049	0.017	No	
IL-15a	0.042	0.013	No	
MCP-1a	0.013	0.010	No	
IL-6a	0.008	0.008	Yes	
IL-1RA	0.007	0.006	Yes	
IL-5b	0.002	0.004	Yes	
IL-8a	0	0.002	Yes	
Notes.

MW Mann–Whitney U comparison

BH Benjamini–Hochberg correction

a Increased in CM.

b Decreased in CM.

Levels of inflammatory biomarkers were lower in survivors than in deceased of CM

Failure to control inflammation is proposed as one of the mechanisms leading to CM, which is consistent with the difference we observed between NCM and CM patients. To further analyze the effect of the inflammatory biomarkers, we compared the levels of analyte between the survivors (n = 18) and the deceased (n = 9) of CM. Interestingly, after multiple testing adjustment (Table 3), there were four biomarkers whose levels significantly differed between the two groups. All were pro-inflammatory analytes (Eotaxin, IL-15, MCP-1 and TNFα) that were significantly lower in the survivors than in the deceased of CM (Fig. 4). These results suggest that the cause of death involved an inflammatory response of high magnitude that was not properly controlled. To analyze possible effects of the inflammatory response in tissue damage, we compared the failure of different organs between survivors and deceased CM patients. Kidney failure was significantly associated with patient’s death (χ2(1, N = 27) = 8.98, p = 0.003; Effect Size = 0.58) while the occurrence of neurological, respiratory, liver, hematologic and hemodynamic failures were comparable between the two groups. We further attempted to correlate the biomarker levels with organ failure. Kidney failure showed significant moderate to strong positive correlations with several chemokines and pro-inflammatory cytokines (Table 4), while respiratory, hematological and liver failure displayed weak positive correlations with 5, 5 and 1 biomarkers respectively (not shown).

Figure 4 Levels of inflammatory immune mediators are higher in deceased than in survivors of cerebral malaria.

The ascendant biomarker profile curve of the survivors (line) was plotted on the deceased graph (bars) to visualize the difference in the proportion of high biomarker producers. Hatched bars represent biomarkers for which there is a significant difference in Mann–Whitney U pairwise comparison between the two groups after Benjamini–Hochberg multiple test adjustment.

Table 3 Biomarkers significantly increased in deceased of cerebral malaria compared to survivors.

Biomarkers	MW P alues	BH critical values	Significance	
IL-6	0.027	0.015	No	
IL-8	0.017	0.010	No	
Eotaxin	0.007	0.008	Yes	
TNFα	0.003	0.006	Yes	
IL-15	0.002	0.004	Yes	
MCP-1	0.001	0.002	Yes	
Notes.

MW Mann–Whitney U comparison

BH Benjamini–Hochberg correction

Table 4 Biomarkers correlated with kidney failure in cerebral malaria patients.

	ρ (p value)		ρ (p value)		ρ (p value)	
Eotaxin	0.514 (0.006)	IL-10	0.457 (0.017)	IL-1α	0.593 (0.001)	
G-CSF	0.500 (0.008)	IL-12p70	0.445 (0.020)	IP-10	0.533 (0.004)	
GM-CSF	0.714 (<0.001)	IL-15	0.621 (0.001)	MCP-1	0.581 (0.002)	
IFNα2	0.525 (0.005)	IL-17A	0.401 (0.038)	TNFα	0.542 (0.003)	
IFNγ	0.529 (0.005)	IL-1RA	0.390 (0.044)			
Notes.

ρ, Spearman’s Rho coefficient.

Variation of biomarker levels before and after cerebral malaria treatment

Analysis of biomarker profiles in malaria patients before and after treatment provides valuable information on immune mediators that are induced during malaria. We compared the levels of biomarker between the time of emergency admission and of hospital release in 14 CM patients (Table 5). Wilcoxon rank test showed 7 biomarkers (G-CSF, IL-10, IL-1α, IL-8, IP-10, MCP-1, TNFα) that were significantly different between the two time points after Benjamini–Hochberg adjustment. All these biomarkers were lower in the second samples confirming the induction of different types of immune mediators including growth factor (G-CSF), inflammatory (TNFα, IL-1α, IP-10), anti-inflammatory (IL-10) and chemokines (IL-8, MCP-1) during immune response to malaria (Table 5).

Table 5 Variation of biomarker levels between admission and release from hospital in cerebral malaria patients.

	Admission (pg/ml) MN ± SD/MD	Release (pg/ml) MN ± SD/MD	P value	
G-CSF	651 ± 1,869/163	116 ± 111/99	0.007	
IL-10	1,848 ±1,749/464	99 ± 165/40	0.002	
TNFα	90 ± 101/60	24 ± 14/20	0.002	
IL-1α	121 ± 165/76	57 ± 80/28	0.013	
IL-8	213 ± 417/66	58 ± 152/14	0.001	
IP-10	5,668 ± 6,297/2,496	1,757 ± 2,191/892	0.002	
MCP-1	1,303 ± 1,754/508	409 ± 678/242	0.005	
Notes.

Admission, biomarker levels at the time of hospital admission of CM patients; Release, biomarker levels at the time of release of CM patients from hospital; P value, two-tailed p value of a Wilcoxon Rank test; MN, mean; SD, standard deviation; MD, median.

Discussion

Inflammation and outcome of cerebral malaria

In this study, we performed a retrospective analysis of 18 controls, and of 17 and 27 NCM and CM patients respectively. The CM patients included 18 survivors and nine (30%) deceased subjects, a proportion similar to the highest mortality rates reported for CM. Beside neurological defect, kidney failure was the most frequent organ malfunction in CM patients and was correlated with death. Analysis of the cytokine response showed a strong induction of pro- and anti-inflammatory biomarkers in malaria patients. However, the magnitude of this response was significantly higher in CM than in NCM patients for inflammatory biomarkers while it was comparable in the two groups for the anti-inflammatory cytokines. Additionally, comparison of the biomarkers in the survivors versus the deceased of CM showed four pro-inflammatory analytes that were significantly higher in the deceased patients. Altogether, our results suggest a scenario in which a strong inflammatory response that was not properly contained led to organ failure and death during CM.

The involvement of the inflammatory response in the pathogenesis of severe malaria, including CM, is well documented (Clark et al., 2008). The balance of pro- and anti-inflammatory cytokines, chemokines and growth factors is key to controlling parasite development without damages to host organs. Regarding the individual biomarkers, we found pro-inflammatory immune mediators increased in malaria patients with levels of IL-6 and IL-8 higher in CM compared to NCM individuals, and of eotaxin, IL-15, MCP-1 and TNFα elevated in deceased compared to survivors of CM. The association of these cytokines and chemokines with malaria severity and/or poor outcome have been described before (Clark et al., 2008). TNFα is one of the first recognized pro-inflammatory biomarkers that play important role during malaria. With other Th1 type cytokines IL-1, IL-12 and IFNγ it contributes to the control of the infection (Schofield & Grau, 2005). However, elevated levels of TNFα were associated with disease severity in both children and adults (Prakash et al., 2006; Thuma et al., 2011) and can discriminate between SM and UM (Mahanta et al., 2015). Similarly, IL-8 (Berg et al., 2014; Lyke et al., 2004), IL-6 (Jakobsen et al., 1994; Lyke et al., 2004), IL-15 (Hu, 2013; Ong’echa et al., 2011) and MCP-1 (MacMullin et al., 2012; Quelhas et al., 2012) were reported as increased during malaria.

In contrast to TNFα, IL-6, IL-8, IL-15 and MCP-1, eotaxin was not often mentioned in previous malaria studies. Interestingly, the level of eotaxin was higher in the CT individuals than in the malaria patients but the difference lost statistical significance after multiple testing adjustment. A significantly lower level of eotaxin was reported by Requena et al. (2014) in pregnant women exposed to malaria when compared to controls residing in malaria-free areas. Additionally, eotaxin was found as a negative predictor of hemoglobin level in children with SMA (Ong’echa et al., 2011). These observations suggest that eotaxin is dowregulated during malaria and that it could be involved in pathogenesis. Eotaxin is a Th2-type chemokine that mediates eosinophil development and recruitment in host tissues (Pope et al., 2001; Queto et al., 2010). Eotaxin is an important biomarker of allergic diseases (Pope et al., 2001) and polymorphism of its encoding gene influence total serum IgE level (Batra et al., 2007; Wang et al., 2007). The role played by IgE in response to malaria infection is controversial with some studies claiming a protective function (Bereczky et al., 2004; Farouk et al., 2005) while other associating IgE with disease severity (Perlmann et al., 1994; Perlmann et al., 1997; Seka-Seka et al., 2004). However, a recent study in a mouse model of experimental CM showed that animals genetically deficient for IgE or for the high affinity receptor for IgE were less susceptible to CM (Porcherie et al., 2011) supporting a role of IgE in the development of CM. The same study showed that CM pathogenesis was mediated by neutrophils expressing the high affinity receptor for IgE that homed to the brain and locally induced high levels of pro-inflammatory cytokines (Porcherie et al., 2011). Whether this function could be translated to human is unknown. However, a recent study reported an elevated neutrophil count that correlated with expression levels of the pro-inflammatory mediators IL-1β and IL-8 in human severe malaria (Mahanta et al., 2015). Altogether, these observations support the hypothesis that elevated levels of eotaxin result in higher production of IgE and deleterious effects during human malaria. If this is the case, the downregulation of eotaxin observed previously (Requena et al., 2014) and in this study might be a mechanism that protects against the damages caused by IgE during malaria. This hypothesis is consistent with higher levels of eotaxin observed in deceased compared to survivors of CM. However it needs to be tested in other studies.

Besides eotaxin, IL-5, another Th2 type cytokine displayed a remarkable profile in this study with its level significantly decreased in CM compared to NCM patients (Fig. 3). IL-5 is a regulatory cytokine that cooperates with eotaxin in the development and recruitment of eosinophils (Nussbaum et al., 2013). A previous study reported elevated levels of IL-5 in mild compared to severe malaria patients (Prakash et al., 2006) suggesting a protective role of this cytokine. This hypothesis is consistent with a recent report of a mouse study demonstrating a protection of rodent against experimental CM by IL-33 treatment (Besnard et al., 2015). The protection against CM was mediated by IL-5 independently of eosinophils, implying a mechanism that does not involve eotaxin.

In conclusion, our study confirms previously reported inflammatory response during malaria. Our findings support the idea of a strong induction of pro-inflammatory immune mediators that was not matched by the production of regulatory, anti-inflammatory biomarkers as the cause of death during CM. Additionally, our results suggests the involvement of eotaxin and of IL-5 in CM development and outcome.

Supplemental Information

File S1 Correlations among biomarkers

Click here for additional data file.

Supplemental Information 1 Raw data: luminex output file

Click here for additional data file.

Supplemental Information 2 Raw data: statistical analyses without children

Click here for additional data file.

Supplemental Information 3 Raw data: statistics on proportion of high levels

Click here for additional data file.

Supplemental Information 4 Raw data: IL-5 ratio analyses

Click here for additional data file.

Supplemental Information 5 Raw data: Mann–Whitney U analysis

Click here for additional data file.

We would like to thank Theresa Wan-Chen Yap for her help in the dosage of the biomarkers. We are grateful to Dr Becaye Fall, Dr Pape Samba Fall and Dr Ronald Perraut for their constructive suggestions and for stimulating discussions.

Additional Information and Declarations

Competing Interests

Author Contributions

Human Ethics

Data Availability

The authors declare there are no competing interests.

Yakhya Dieye conceived and designed the experiments, performed the experiments, analyzed the data, wrote the paper, prepared figures and/or tables.

Babacar Mbengue conceived and designed the experiments, performed the experiments, analyzed the data, reviewed drafts of the paper.

Shobha Dagamajalu performed the experiments.

Mouhamadou Mansour Fall diagnosis of malaria, determination of blood parameters, treatment of patients.

Mun Fai Loke contributed reagents/materials/analysis tools.

Cheikh Momar Nguer analyzed the data, prepared figures and/or tables.

Alassane Thiam performed the experiments, prepared figures and/or tables.

Jamuna Vadivelu contributed reagents/materials/analysis tools, reviewed drafts of the paper.

Alioune Dieye conceived and designed the experiments, contributed reagents/materials/analysis tools, reviewed drafts of the paper.

The following information was supplied relating to ethical approvals (i.e., approving body and any reference numbers):

Université Cheikh Anta Diop de Dakar’s institutional research ethics committee; Protocol Number 001/2015/CER/UCAD.

The following information was supplied regarding data availability:

The raw data has been supplied as a Supplemental Dataset.

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
