# Peer review of "Cytokine response during non-cerebral and cerebral malaria: evidence of a failure to control inflammation as a cause of death in African adults"

_PeerJ, doi:10.7717/peerj.1965_

## Round 0.1 · original submission · Major Revisions

· Academic Editor

Major Revisions

We believe that the manuscript adds important information to the field and may be considered for publication. Please, consider all the points raised by the two experts reviewers as detailed below. Please, consider to adapt the discussion to make it clearer and more focused. Please, correct typing errors across the main text.

Reviewer 1 ·

Basic reporting

Overall the manuscript is clear and well written. There are a couple of words written in capital letters such as Malaria or Kidney that needs to be written in lowercase.

In the introduction after the immunoepidemiology explanation of malaria it could be added that the fact that non-immune adults are susceptible to severe malaria and cerebral malaria in particular is very important in the context of malaria eradication. The decrease on malaria transmission intensity will affect immunity of adults in endemic areas, rendering them more susceptible to severe malaria.

In the introduction it could be considered to include data on the recently published WHO report instead of the WHO report from 2014.

The second part of the discussion needs some more clarity and could benefit of further comparison of findings with previously reported results. It is not clear what are the contributions of their results to the knowledge on biomarkers of severe and cerebral malaria. Do the biomarkers detected, besides an overall proinflammatory response, confirm previous results or not? and what are the differences between different studies that could explain the variability observed? There is an attempt to do this comparison at the end of the discussion, but it is incomplete and a bit confusing.

Th1 cytokines have been involved in malaria protection and in some manuscripts lower levels of some Th1 cytokines such as IL-12 have been associated with severity. What are the findings regarding the Th1 cytokines and how does it relate to previously published studies?

The finding of lower levels of IL5 associated with CM is interesting. Is the first time it is reported in the literature? It would be of interest to include some hypothesis related to this result in the discussion.

Figures are relevant and of good quality. The only thing that is confusing is that the statistical significance reported is in relation to differences in biomarker levels and not in relation to differences in proportion of individuals with high responses. I would clarify this and show significance on the proportion of responses. In addition I would add figures (box plots ideally with dot plots) on the analyte levels of the different groups tested. It would be very informative for the readers to see the magnitude of the responses besides the percentage of individuals with high biomarker level. Also it may help show the quality of the data used for the analysis.

In several occasions the term “hormones” is used to refer to cytokine/chemokine/growth factor. It would be better to use the specific term of the molecules such as cytokine or chemokine or growth factor, depending on what the sentence is referring to. Similarly, a general term such as “analytes” or “biomarkers” should be used for overall analytes that include the three different types. Sometimes “cytokines” is used in reference to chemokines or growth factors.

Regarding the raw data of biomarker concentrations supplied, there is no data for EGF. What is the meaning of the different colors of the data set?

Experimental design

It seems that there is a mistake in the CM sample size in the text. According to the text 34 subjects had CM whereas in the table there is information only for 27 subjects.

It is not clear what is the definition of cerebral malaria used in the study. In methods it is said that life-threatening CM was characterized by at least one of the criteria defined by the WHO including ... severe anemia, respiratory distress.... However, some of these are not specific for malaria, but for severe malaria.

Does the malaria treatment explained in methods correspond to only CM patients?

Information on biomarker measurement is lacking. Was the serum diluted? What quantity was used? Were all samples tested in a single plate? If more than one plate was used, how was the sample distribution? Luminex assays can have high variability. Were controls used? What is the coefficient of variation of the assay? What models were used for fitting the data and extrapolate from MFI to concentrations? How many samples or proportion of samples had no detectable levels of cytokines or MFI over the upper limit of quantification? How were these values treated and used in the posterior analysis?

Regarding statistical analysis, multiple testing adjustments when using so many variables is highly recommended. Both raw p-values and adjusted p-values should be reported.

Even if there are no differences in parasitemia between NCM and CM it is important to report the parasitemias and it could be included in Table 1.

It is known that cytokine and chemokine responses vary with age and the profile of children may be different than adults. In addition, all children are in the NCM. If they are no excluded in the analysis, its potential confounding effect on the analyte concentrations should be discussed.

Significant correlations were observed between several biomarkers, but only the correlation between TNFa and IL-10 is shown. I think it is of interest to show all correlations as well as if there are differences in correlations depending on the malaria group. In addition, correlations with parasitemia stratified by malaria group too may also be very informative. Differences in these correlations may reflect differential regulatory mechanisms.

The correlation test to analyse the effect of biomarkers in organ failure in CM does not seem to be the adequate statistical test since organ failure variables are categorical with only two levels. A comparison of biomarker level could be performed between the two groups or maybe a logistic regression with the organ failure as a dependent variable.

Validity of the findings

The conclusion of the manuscript is supported by the findings, but lack of information on the biomarkers detection and analysis methods and other concerns raised (explained in the comments above) make it difficult to know if data is robust and statistically sound.

Reviewer 2 ·

Basic reporting

The article is well written, although there a a few typographical and grammatical errors as well as omissions that need to be corrected. The literature that has been cited is relevant and adequate. The following errors were identified;
Abstract, line 32 - ".... suffering from ...." instead of ".....suffering of ..."
Abstract, line 29 & Results line 160 - should this be "..... cytokines/inflammatory mediators....." instead of "... hormones..." ?
Introduction, line 58 - delete "to"
Introduction, line 63 - "cause" instead of "causes"
Introduction, lines 82 and 83 - consider revising sentence for clarity
Methods, line 103 - "consisted of" rather than "consisted to "
Methods, line 127 - " p value < 0.05", not "0.5"
Results, lines 187/188 - ...... occurrence "of" neurological........
Discussion, line 242 - delete "a"
Discussion, lines 248 to the end - font size seems different
Discussion, lines 254 - 260 - sentence seems too long, consider revising
Figures 1A and 1B, two of the interleukins (IL-) on the horizontal axis are missing the numbers

Experimental design

The main experimental work was the running of samples by the Luminex xMAP technology and reference for the actual procedure has only been made to the kit manufacturer's recommendations.
It is unclear why authors decided to define a cut off based on the median of the entire study population, rather than the norm of using the data from healthy controls.

Validity of the findings

The findings of the study as reported are valid. It may however be important for the authors to also show data on ratios between relevant pro- and anti-inflammatory cytokines as was done by Kurtzhal's et al (1998) since this and other published works clearly show that it is not just a matter of high or low levels of pro- and anti-inflammatory cyokines that matter, but the balance between the two groups of cytokines.
There is no conclusion highlighting the key findings of the study

---

## Round 0.2 · Minor Revisions

· Academic Editor

Minor Revisions

The manuscript was properly revised but a few minor issues require attention as detailed by the two reviewers.

Reviewer 1 ·

Basic reporting

Methods (lines 141-144): consider revising the paragraph for clarity.
Discussion (line 280): A significantly lower, not higher, level of eotaxin was reported by Requena et al. in exposed pregnant women.
Consider adding a reference to the discussion: Rovira-Vallbona et al. 2012. They reported an imbalanced pro-inflammatory response in severe malaria compared to uncomplicated malaria (lower levels of IL-12 and the immunoregulatory TGFb and higher levels of IL-6).

Experimental design

No comments.

Validity of the findings

The hypothesis of an eotaxin-IgE mechanism and the suggestion of down-regulation of eotaxin as a mechanisms of protection against malaria severity is highly speculative, which is fine, but it is not clear how it would fit with the Requena et al. findings of eotaxin correlating with different B cell populations and IgG levels.

Additional comments

The authors have satisfactory addressed all previous concerns and the manuscript has improved in clarity and quality.

Reviewer 2 ·

Basic reporting

line 118 (of the tracked changes version of the manuscript) - .......no other cause of cerebral symptoms
lines 15 - 159 - the sentence is not very clear. Please clarify, especially line 158

Experimental design

No comments

Validity of the findings

No comments

External reviews were received for this submission. These reviews were used by the Editor when they made their decision, and can be downloaded below.

---

## Round 0.3 · accepted · Accept

· Academic Editor

Accept

All the points raised by the reviewers have been properly addressed.